# Escape saddle points by a simple gradient-descent based algorithm

**Chenyi Zhang**[1]     **Tongyang Li**[2,3,4*]

[1] Institute for Interdisciplinary Information Sciences, Tsinghua University, China
[2] Center on Frontiers of Computing Studies, Peking University, China
[3] School of Computer Science, Peking University, China
[4] Center for Theoretical Physics, Massachusetts Institute of Technology, USA

## Abstract

Escaping saddle points is a central research topic in nonconvex optimization. In this paper, we propose a simple gradient-based algorithm such that for a smooth function $f \colon \mathbb{R}^n \to \mathbb{R}$, it outputs an $\epsilon$-approximate second-order stationary point in $\tilde{O}(\log n/\epsilon^{1.75})$ iterations. Compared to the previous state-of-the-art algorithms by Jin et al. with $\tilde{O}(\log^4 n/\epsilon^2)$ or $\tilde{O}(\log^6 n/\epsilon^{1.75})$ iterations, our algorithm is polynomially better in terms of $\log n$ and matches their complexities in terms of $1/\epsilon$. For the stochastic setting, our algorithm outputs an $\epsilon$-approximate second-order stationary point in $\tilde{O}(\log^2 n/\epsilon^4)$ iterations. Technically, our main contribution is an idea of implementing a robust Hessian power method using only gradients, which can find negative curvature near saddle points and achieve the polynomial speedup in $\log n$ compared to the perturbed gradient descent methods. Finally, we also perform numerical experiments that support our results.

## 1   Introduction

Nonconvex optimization is a central research area in optimization theory, since lots of modern machine learning problems can be formulated in models with nonconvex loss functions, including deep neural networks, principal component analysis, tensor decomposition, etc. In general, finding a global minimum of a nonconvex function is NP-hard in the worst case. Instead, many theoretical works focus on finding a local minimum instead of a global one, because recent works (both empirical and theoretical) suggested that local minima are nearly as good as global minima for a significant amount of well-studied machine learning problems; see e.g. [4, 12, 14, 15, 17, 18]. On the other hand, saddle points are major obstacles for solving these problems, not only because they are ubiquitous in high-dimensional settings where the directions for escaping may be few (see e.g. [5, 8, 11]), but also saddle points can correspond to highly suboptimal solutions (see e.g. [19, 28]).

Hence, one of the most important topics in nonconvex optimization is to *escape saddle points*. Specifically, we consider a twice-differentiable function $f \colon \mathbb{R}^n \to \mathbb{R}$ such that

- $f$ is $\ell$-smooth: $\|\nabla f(\mathbf{x}_1) - \nabla f(\mathbf{x}_2)\| \le \ell \|\mathbf{x}_1 - \mathbf{x}_2\|$   $\forall \mathbf{x}_1, \mathbf{x}_2 \in \mathbb{R}^n$,
- $f$ is $\rho$-Hessian Lipschitz: $\|\mathcal{H}(\mathbf{x}_1) - \mathcal{H}(\mathbf{x}_2)\| \le \rho \|\mathbf{x}_1 - \mathbf{x}_2\|$   $\forall \mathbf{x}_1, \mathbf{x}_2 \in \mathbb{R}^n$;

here $\mathcal{H}$ is the Hessian of $f$. The goal is to find an $\epsilon$-approximate second-order stationary point $\mathbf{x}_\epsilon$:[1]

$$\|\nabla f(\mathbf{x}_\epsilon)\| \le \epsilon, \quad \lambda_{\min}(\mathcal{H}(\mathbf{x}_\epsilon)) \ge -\sqrt{\rho\epsilon}. \tag{1}$$

---

*Corresponding author. Email: tongyangli@pku.edu.cn

[1]We can ask for an $(\epsilon_1, \epsilon_2)$-approx. second-order stationary point s.t. $\|\nabla f(\mathbf{x})\| \le \epsilon_1$ and $\lambda_{\min}(\nabla^2 f(\mathbf{x})) \ge -\epsilon_2$ in general. The scaling in (1) was adopted as a standard in literature [1, 6, 10, 20, 21, 22, 26, 29, 30, 31].

35th Conference on Neural Information Processing Systems (NeurIPS 2021).

In other words, at any $\epsilon$-approx. second-order stationary point $\mathbf{x}_\epsilon$, the gradient is small with norm being at most $\epsilon$ and the Hessian is close to be positive semi-definite with all its eigenvalues $\geq -\sqrt{\rho\epsilon}$.

Algorithms for escaping saddle points are mainly evaluated from two aspects. On the one hand, considering the enormous dimensions of machine learning models in practice, dimension-free or almost dimension-free (i.e., having $\text{poly}(\log n)$ dependence) algorithms are highly preferred. On the other hand, recent empirical discoveries in machine learning suggests that it is often feasible to tackle difficult real-world problems using simple algorithms, which can be implemented and maintained more easily in practice. On the contrary, algorithms with nested loops often suffer from significant overheads in large scales, or introduce concerns with the setting of hyperparameters and numerical stability (see e.g. [1, 7]), making them relatively hard to find practical implementations.

It is then natural to explore simple gradient-based algorithms for escaping from saddle points. The reason we do not assume access to Hessians is because its construction takes $\Omega(n^2)$ cost in general, which is computationally infeasible when the dimension is large. A seminal work along this line was by Ge et al. [12], which found an $\epsilon$-approximate second-order stationary point satisfying (1) using only gradients in $O(\text{poly}(n, 1/\epsilon))$ iterations. This is later improved to be almost dimension-free $\tilde{O}(\log^4 n/\epsilon^2)$ in the follow-up work [20],[2] and the perturbed accelerated gradient descent algorithm [22] based on Nesterov's accelerated gradient descent [27] takes $\tilde{O}(\log^6 n/\epsilon^{1.75})$ iterations. However, these results still suffer from a significant overhead in terms of $\log n$. On the other direction, Refs. [3, 25, 30] demonstrate that an $\epsilon$-approximate second-order stationary point can be find using gradients in $\tilde{O}(\log n/\epsilon^{1.75})$ iterations. Their results are based on previous works [1, 6] using Hessian-vector products and the observation that the Hessian-vector product can be approximated via the difference of two gradient queries. Hence, their implementations contain nested-loop structures with relatively large numbers of hyperparameters. It has been an open question whether it is possible to keep both the merits of using only first-order information as well as being close to dimension-free using a simple, gradient-based algorithm without a nested-loop structure [23]. This paper answers this question in the affirmative.

**Contributions.** Our main contribution is a simple, single-loop, and robust gradient-based algorithm that can find an $\epsilon$-approximate second-order stationary point of a smooth, Hessian Lipschitz function $f: \mathbb{R}^n \to \mathbb{R}$. Compared to previous works [3, 25, 30] exploiting the idea of gradient-based Hessian power method, our algorithm has a single-looped, simpler structure and better numerical stability. Compared to the previous state-of-the-art results with single-looped structures by [22] and [20, 21] using $\tilde{O}(\log^6 n/\epsilon^{1.75})$ or $\tilde{O}(\log^4 n/\epsilon^2)$ iterations, our algorithm achieves a polynomial speedup in $\log n$:

**Theorem 1** (informal). *Our single-looped algorithm finds an $\epsilon$-approximate second-order stationary point in $\tilde{O}(\log n/\epsilon^{1.75})$ iterations.*

Technically, our work is inspired by the perturbed gradient descent (PGD) algorithm in [20, 21] and the perturbed accelerated gradient descent (PAGD) algorithm in [22]. Specifically, PGD applies gradient descents iteratively until it reaches a point with small gradient, which can be a potential saddle point. Then PGD generates a uniform perturbation in a small ball centered at that point and then continues the GD procedure. It is demonstrated that, with an appropriate choice of the perturbation radius, PGD can shake the point off from the neighborhood of the saddle point and converge to a second-order stationary point with high probability. The PAGD in [22] adopts a similar perturbation idea, but the GD is replaced by Nesterov's AGD [27].

Our algorithm is built upon PGD and PAGD but with one main modification regarding the perturbation idea: it is more efficient to add a perturbation in the *negative curvature direction* nearby the saddle point, rather than the uniform perturbation in PGD and PAGD, which is a compromise since we generally cannot access the Hessian at the saddle due to its high computational cost. Our key observation lies in the fact that we do not have to compute the entire Hessian to detect the negative curvature. Instead, in a small neighborhood of a saddle point, gradients can be viewed as Hessian-vector products plus some bounded deviation. In particular, GD near the saddle with learning rate $1/\ell$ is approximately the same as the power method of the matrix $(I - \mathcal{H}/\ell)$. As a result, the most negative eigenvalues stand out in GD because they have leading exponents in the power method, and thus it approximately moves along the direction of the most negative curvature nearby the sad-

---

[2] The $\tilde{O}$ notation omits poly-logarithmic terms, i.e., $\tilde{O}(g) = O(g \, \text{poly}(\log g))$.

dle point. Following this approach, we can escape the saddle points more rapidly than previous algorithms: for a constant $\epsilon$, PGD and PAGD take $O(\log n)$ iterations to decrease the function value by $\Omega(1/\log^3 n)$ and $\Omega(1/\log^5 n)$ with high probability, respectively; on the contrary, we can first take $O(\log n)$ iterations to specify a negative curvature direction, and then add a larger perturbation in this direction to decrease the function value by $\Omega(1)$. See Proposition 3 and Proposition 5. After escaping the saddle point, similar to PGD and PAGD, we switch back to GD and AGD iterations, which are efficient to decrease the function value when the gradient is large [20, 21, 22].

Our algorithm is also applicable to the stochastic setting where we can only access stochastic gradients, and the stochasticity is not under the control of our algorithm. We further assume that the stochastic gradients are Lipschitz (or equivalently, the underlying functions are gradient-Lipschitz, see Assumption 2), which is also adopted in most of the existing works; see e.g. [9, 20, 21, 35]. We demonstrate that a simple extended version of our algorithm takes $O(\log^2 n)$ iterations to detect a negative curvature direction using only stochastic gradients, and then obtain an $\Omega(1)$ function value decrease with high probability. On the contrary, the perturbed stochastic gradient descent (PSGD) algorithm in [20, 21], the stochastic version of PGD, takes $O(\log^{10} n)$ iterations to decrease the function value by $\Omega(1/\log^5 n)$ with high probability.

**Theorem 2** (informal). *In the stochastic setting, our algorithm finds an $\epsilon$-approximate second-order stationary point using $\tilde{O}(\log^2 n/\epsilon^4)$ iterations via stochastic gradients.*

Our results are summarized in Table 1. Although the underlying dynamics in [3, 25, 30] and our algorithm have similarity, the main focus of our work is different. Specifically, Refs. [3, 25, 30] mainly aim at using novel techniques to reduce the iteration complexity for finding a second-order stationary point, whereas our work mainly focuses on reducing the number of loops and hyper-parameters of negative curvature finding methods while preserving their advantage in iteration complexity, since a much simpler structure accords with empirical observations and enables wider applications. Moreover, the choice of perturbation in [3] is based on the Chebyshev approximation theory, which may require additional nested-looped structures to boost the success probability. In the stochastic setting, there are also other results studying nonconvex optimization [16, 24, 32, 37, 13, 33, 36] from different perspectives than escaping saddle points, which are incomparable to our results.

| Setting | Reference | Oracle | Iterations | Simplicity |
|---|---|---|---|---|
| Non-stochastic | [1, 6] | Hessian-vector product | $\tilde{O}(\log n/\epsilon^{1.75})$ | Nested-loop |
| Non-stochastic | [20, 21] | Gradient | $\tilde{O}(\log^4 n/\epsilon^2)$ | Single-loop |
| Non-stochastic | [22] | Gradient | $\tilde{O}(\log^6 n/\epsilon^{1.75})$ | Single-loop |
| Non-stochastic | [3, 25, 30] | Gradient | $\tilde{O}(\log n/\epsilon^{1.75})$ | Nested-loop |
| Non-stochastic | **this work** | Gradient | $\tilde{O}(\log n/\epsilon^{1.75})$ | Single-loop |
| Stochastic | [20, 21] | Gradient | $\tilde{O}(\log^{15} n/\epsilon^4)$ | Single-loop |
| Stochastic | [10] | Gradient | $\tilde{O}(\log^5 n/\epsilon^{3.5})$ | Single-loop |
| Stochastic | [3] | Gradient | $\tilde{O}(\log^2 n/\epsilon^{3.5})$ | Nested-loop |
| Stochastic | [9] | Gradient | $\tilde{O}(\log^2 n/\epsilon^3)$ | Nested-loop |
| Stochastic | **this work** | Gradient | $\tilde{O}(\log^2 n/\epsilon^4)$ | Single-loop |

Table 1: A summary of the state-of-the-art results on finding approximate second-order stationary points by the first-order (gradient) oracle. Iteration numbers are highlighted in terms of the dimension $n$ and the precision $\epsilon$.

It is worth highlighting that our gradient-descent based algorithm enjoys the following nice features:

- *Simplicity:* Some of the previous algorithms have nested-loop structures with the concern of practical impact when setting the hyperparameters. In contrast, our algorithm based on negative curvature finding only contains a single loop with two components: gradient descent (including AGD or SGD) and perturbation. As mentioned above, such simple structure is preferred in machine learning, which increases the possibility of our algorithm to find real-world applications.

- *Numerical stability:* Our algorithm contains an additional renormalization step at each iteration when escaping from saddle points. Although in theoretical aspect a renormalization step does not affect the output and the complexity of our algorithm, when finding negative curvature near

saddle points it enables us to sample gradients in a larger region, which makes our algorithm more numerically stable against floating point error and other errors. The introduction of renormalization step is enabled by the simple structure of our algorithm, which may not be feasible for more complicated algorithms [3, 25, 30].

- *Robustness:* Our algorithm is robust against adversarial attacks when evaluating the value of the gradient. Specifically, when analyzing the performance of our algorithm near saddle points, we essentially view the deflation from pure quadratic geometry as an external noise. Hence, the effectiveness of our algorithm is unaffected under external attacks as long as the adversary is bounded by deflations from quadratic landscape.

Finally, we perform numerical experiments that support our polynomial speedup in $\log n$. We perform our negative curvature finding algorithms using GD or SGD in various landscapes and general classes of nonconvex functions, and use comparative studies to show that our Algorithm 1 and Algorithm 3 achieve a higher probability of escaping saddle points using much fewer iterations than PGD and PSGD (typically less than $1/3$ times of the iteration number of PGD and $1/2$ times of the iteration number of PSGD, respectively). Moreover, we perform numerical experiments benchmarking the solution quality and iteration complexity of our algorithm against accelerated methods. Compared to PAGD [22] and even advanced optimization algorithms such as NEON+ [30], Algorithm 2 possesses better solution quality and iteration complexity in various landscapes given by more general nonconvex functions. With fewer iterations compared to PAGD and NEON+ (typically less than $1/3$ times of the iteration number of PAGD and $1/2$ times of the iteration number of NEON+, respectively), our Algorithm 2 achieves a higher probability of escaping from saddle points.

**Open questions.** This work leaves a couple of natural open questions for future investigation:

- Can we achieve the polynomial speedup in $\log n$ for more advanced stochastic optimization algorithms with complexity $\tilde{O}(\text{poly}(\log n)/\epsilon^{3.5})$ [2, 3, 10, 29, 31] or $\tilde{O}(\text{poly}(\log n)/\epsilon^3)$ [9, 34]?
- How is the performance of our algorithms for escaping saddle points in real-world applications, such as tensor decomposition [12, 17], matrix completion [14], etc.?

**Broader impact.** This work focuses on the theory of nonconvex optimization, and as far as we see, we do not anticipate its potential negative societal impact. Nevertheless, it might have a positive impact for researchers who are interested in understanding the theoretical underpinnings of (stochastic) gradient descent methods for machine learning applications.

**Organization.** In Section 2, we introduce our gradient-based Hessian power method algorithm for negative curvature finding, and present how our algorithms provide polynomial speedup in $\log n$ for both PGD and PAGD. In Section 3, we present the stochastic version of our negative curvature finding algorithm using stochastic gradients and demonstrate its polynomial speedup in $\log n$ for PSGD. Numerical experiments are presented in Section 4. We provide detailed proofs and additional numerical experiments in the supplementary material.

## 2 A simple algorithm for negative curvature finding

We show how to find negative curvature near a saddle point using a gradient-based Hessian power method algorithm, and extend it to a version with faster convergence rate by replacing gradient descents by accelerated gradient descents. The intuition works as follows: in a small enough region nearby a saddle point, the gradient can be approximately expressed as a Hessian-vector product formula, and the approximation error can be efficiently upper bounded, see Eq. (6). Hence, using only gradients information, we can implement an accurate enough Hessian power method to find negative eigenvectors of the Hessian matrix, and further find the negative curvature nearby the saddle.

### 2.1 Negative curvature finding based on gradient descents

We first present an algorithm for negative curvature finding based on gradient descents. Specifically, for any $\tilde{\mathbf{x}} \in \mathbb{R}^n$ with $\lambda_{\min}(\mathcal{H}(\tilde{\mathbf{x}})) \leq -\sqrt{\rho\epsilon}$, it finds a unit vector $\hat{\mathbf{e}}$ such that $\hat{\mathbf{e}}^T \mathcal{H}(\tilde{\mathbf{x}})\hat{\mathbf{e}} \leq -\sqrt{\rho\epsilon}/4$.

---

**Algorithm 1:** Negative Curvature Finding($\tilde{\mathbf{x}}, r, \mathscr{T}$).

---

**1** $\mathbf{y}_0 \leftarrow \text{Uniform}(\mathbb{B}_{\tilde{\mathbf{x}}}(r))$ where $\mathbb{B}_{\tilde{\mathbf{x}}}(r)$ is the $\ell_2$-norm ball centered at $\tilde{\mathbf{x}}$ with radius $r$;
**2 for** $t = 1, ..., \mathscr{T}$ **do**
**3** $\quad \big\lfloor \quad \mathbf{y}_t \leftarrow \mathbf{y}_{t-1} - \frac{\|\mathbf{y}_{t-1}\|}{\ell r}\big(\nabla f(\tilde{\mathbf{x}} + r\mathbf{y}_{t-1}/\|\mathbf{y}_{t-1}\|) - \nabla f(\tilde{\mathbf{x}})\big)$ ;
**4 Output** $\mathbf{y}_{\mathscr{T}}/r$.

---

**Proposition 3.** *Suppose the function $f \colon \mathbb{R}^n \to \mathbb{R}$ is $\ell$-smooth and $\rho$-Hessian Lipschitz. For any $0 < \delta_0 \leq 1$, we specify our choice of parameters and constants we use as follows:*

$$\mathscr{T} = \frac{8\ell}{\sqrt{\rho\epsilon}} \cdot \log\Big(\frac{\ell}{\delta_0}\sqrt{\frac{n}{\pi\rho\epsilon}}\Big), \quad r = \frac{\epsilon}{8\ell}\sqrt{\frac{\pi}{n}}\delta_0. \tag{2}$$

*Suppose that $\tilde{\mathbf{x}}$ satisfies $\lambda_{\min}(\nabla^2 f(\tilde{\mathbf{x}})) \leq -\sqrt{\rho\epsilon}$. Then with probability at least $1 - \delta_0$, Algorithm 1 outputs a unit vector $\hat{\mathbf{e}}$ satisfying*

$$\hat{\mathbf{e}}^T \mathcal{H}(\mathbf{x})\hat{\mathbf{e}} \leq -\sqrt{\rho\epsilon}/4, \tag{3}$$

*using $O(\mathscr{T}) = \tilde{O}\Big(\frac{\log n}{\sqrt{\rho\epsilon}}\Big)$ iterations, where $\mathcal{H}$ stands for the Hessian matrix of function $f$.*

*Proof.* Without loss of generality we assume $\tilde{\mathbf{x}} = \mathbf{0}$ by shifting $\mathbb{R}^n$ such that $\tilde{\mathbf{x}}$ is mapped to $\mathbf{0}$. Define a new $n$-dimensional function

$$h_f(\mathbf{x}) := f(\mathbf{x}) - \langle \nabla f(\mathbf{0}), \mathbf{x} \rangle, \tag{4}$$

for the ease of our analysis. Since $\langle \nabla f(\mathbf{0}), \mathbf{x} \rangle$ is a linear function with Hessian being 0, the Hessian of $h_f$ equals to the Hessian of $f$, and $h_f(\mathbf{x})$ is also $\ell$-smooth and $\rho$-Hessian Lipschitz. In addition, note that $\nabla h_f(\mathbf{0}) = \nabla f(\mathbf{0}) - \nabla f(\mathbf{0}) = 0$. Then for all $\mathbf{x} \in \mathbb{R}^n$,

$$\nabla h_f(\mathbf{x}) = \int_{\xi=0}^1 \mathcal{H}(\xi\mathbf{x}) \cdot \mathbf{x}\,\mathrm{d}\xi = \mathcal{H}(\mathbf{0})\mathbf{x} + \int_{\xi=0}^1 (\mathcal{H}(\xi\mathbf{x}) - \mathcal{H}(\mathbf{0})) \cdot \mathbf{x}\,\mathrm{d}\xi. \tag{5}$$

Furthermore, due to the $\rho$-Hessian Lipschitz condition of both $f$ and $h_f$, for any $\xi \in [0, 1]$ we have $\|\mathcal{H}(\xi\mathbf{x}) - \mathcal{H}(\mathbf{0})\| \leq \rho\|\mathbf{x}\|$, which leads to

$$\|\nabla h_f(\mathbf{x}) - \mathcal{H}(\mathbf{0})\mathbf{x}\| \leq \rho\|\mathbf{x}\|^2. \tag{6}$$

Observe that the Hessian matrix $\mathcal{H}(\mathbf{0})$ admits the following eigen-decomposition:

$$\mathcal{H}(\mathbf{0}) = \sum_{i=1}^n \lambda_i \mathbf{u}_i \mathbf{u}_i^T, \tag{7}$$

where the set $\{\mathbf{u}_i\}_{i=1}^n$ forms an orthonormal basis of $\mathbb{R}^n$. Without loss of generality, we assume the eigenvalues $\lambda_1, \lambda_2, \ldots, \lambda_n$ corresponding to $\mathbf{u}_1, \mathbf{u}_2, \ldots, \mathbf{u}_n$ satisfy

$$\lambda_1 \leq \lambda_2 \leq \cdots \leq \lambda_n, \tag{8}$$

in which $\lambda_1 \leq -\sqrt{\rho\epsilon}$. If $\lambda_n \leq -\sqrt{\rho\epsilon}/2$, Proposition 3 holds directly. Hence, we only need to prove the case where $\lambda_n > -\sqrt{\rho\epsilon}/2$, in which there exists some $p, p'$ with

$$\lambda_p \leq -\sqrt{\rho\epsilon} < \lambda_{p+1}, \quad \lambda_{p'} \leq -\sqrt{\rho\epsilon}/2 < \lambda_{p'+1}. \tag{9}$$

We use $\mathfrak{S}_\parallel$, $\mathfrak{S}_\perp$ to separately denote the subspace of $\mathbb{R}^n$ spanned by $\{\mathbf{u}_1, \mathbf{u}_2, \ldots, \mathbf{u}_p\}$, $\{\mathbf{u}_{p+1}, \mathbf{u}_{p+2}, \ldots, \mathbf{u}_n\}$, and use $\mathfrak{S}'_\parallel$, $\mathfrak{S}'_\perp$ to denote the subspace of $\mathbb{R}^n$ spanned by $\{\mathbf{u}_1, \mathbf{u}_2, \ldots, \mathbf{u}_{p'}\}$, $\{\mathbf{u}_{p'+1}, \mathbf{u}_{p'+2}, \ldots, \mathbf{u}_n\}$. Furthermore, we define $\mathbf{y}_{t,\parallel} := \sum_{i=1}^p \langle \mathbf{u}_i, \mathbf{y}_t \rangle \mathbf{u}_i$, $\mathbf{y}_{t,\perp} := \sum_{i=p}^n \langle \mathbf{u}_i, \mathbf{y}_t \rangle \mathbf{u}_i$, $\mathbf{y}_{t,\parallel'} := \sum_{i=1}^{p'} \langle \mathbf{u}_i, \mathbf{y}_t \rangle \mathbf{u}_i$, $\mathbf{y}_{t,\perp'} := \sum_{i=p'}^n \langle \mathbf{u}_i, \mathbf{y}_t \rangle \mathbf{u}_i$ respectively to denote the component of $\mathbf{y}_t$ in Line 3 in the subspaces $\mathfrak{S}_\parallel, \mathfrak{S}_\perp, \mathfrak{S}'_\parallel, \mathfrak{S}'_\perp$, and let $\alpha_t := \|\mathbf{y}_{t,\parallel}\|/\|\mathbf{y}_t\|$. Observe that

$$\Pr\Big\{\alpha_0 \geq \delta_0\sqrt{\pi/n}\Big\} \geq \Pr\Big\{|y_{0,1}|/r \geq \delta_0\sqrt{\pi/n}\Big\}, \tag{10}$$

where $y_{0,1} := \langle \mathbf{u}_1, \mathbf{y}_0 \rangle$ denotes the component of $\mathbf{y}_0$ along $\mathbf{u}_1$. Consider the case where $\alpha_0 \geq \delta_0 \sqrt{\pi/n}$, which can be achieved with probability

$$\Pr\left\{\alpha_0 \geq \sqrt{\frac{\pi}{n}}\delta_0\right\} \geq 1 - \sqrt{\frac{\pi}{n}}\delta_0 \cdot \frac{\mathrm{Vol}(\mathbb{B}_0^{n-1}(1))}{\mathrm{Vol}(\mathbb{B}_0^n(1))} \geq 1 - \sqrt{\frac{\pi}{n}}\delta_0 \cdot \sqrt{\frac{n}{\pi}} = 1 - \delta_0. \quad (11)$$

We prove that there exists some $t_0$ with $1 \leq t_0 \leq \mathscr{T}$ such that

$$\|\mathbf{y}_{t_0,\perp'}\|/\|\mathbf{y}_{t_0}\| \leq \sqrt{\rho\epsilon}/(8\ell). \quad (12)$$

Assume the contrary, for any $1 \leq t \leq \mathscr{T}$, we all have $\|\mathbf{y}_{t,\perp'}\|/\|\mathbf{y}_t\| > \sqrt{\rho\epsilon}/(8\ell)$. Then $\|\mathbf{y}'_{t,\perp}\|$ satisfies the following recurrence formula:

$$\|\mathbf{y}_{t+1,\perp'}\| \leq (1 + \sqrt{\rho\epsilon}/(2\ell))\|\mathbf{y}_{t,\perp'}\| + \|\Delta_{\perp'}\| \leq (1 + \sqrt{\rho\epsilon}/(2\ell) + \|\Delta\|/\|\mathbf{y}_{t,\perp'}\|)\|\mathbf{y}_{t,\perp'}\|, \quad (13)$$

where $\Delta := \frac{\|\mathbf{y}_t\|}{r\ell}(\nabla h_f(r\mathbf{y}_t/\|\mathbf{y}_t\|) - \mathcal{H}(0) \cdot (r\mathbf{y}_t/\|\mathbf{y}_t\|))$ and $\|\Delta\|/\|\mathbf{y}_t\| \leq \rho r/\ell$ due to (6). Hence,

$$\|\mathbf{y}_{t+1,\perp'}\| \leq \left(1 + \frac{\sqrt{\rho\epsilon}}{2\ell} + \frac{\|\Delta\|}{\|\mathbf{y}_{t,\perp'}\|}\right)\|\mathbf{y}_{t,\perp'}\| \leq \left(1 + \frac{\sqrt{\rho\epsilon}}{2\ell} + \cdot\frac{8\rho r}{\sqrt{\rho\epsilon}}\right)\|\mathbf{y}_{t+1,\perp'}\|, \quad (14)$$

which leads to

$$\|\mathbf{y}_{t,\perp'}\| \leq \|\mathbf{y}_{0,\perp'}\|(1 + \sqrt{\rho\epsilon}/(2\ell) + 8\rho r/\sqrt{\rho\epsilon})^t \leq \|\mathbf{y}_{0,\perp'}\|(1 + 5\sqrt{\rho\epsilon}/(8\ell))^t, \quad \forall t \in [\mathscr{T}]. \quad (15)$$

Similarly, we can have the recurrence formula for $\|\mathbf{y}_{t,\parallel}\|$:

$$\|\mathbf{y}_{t+1,\parallel}\| \geq (1 + \sqrt{\rho\epsilon}/(2\ell))\|\mathbf{y}_{t,\parallel}\| - \|\Delta_\parallel\| \geq (1 + \sqrt{\rho\epsilon}/(2\ell) - \|\Delta\|/(\alpha_t\|\mathbf{y}_t\|))\|\mathbf{y}_{t,\parallel}\|. \quad (16)$$

Considering that $\|\Delta\|/\|\mathbf{y}_t\| \leq \rho r/\ell$ due to (6), we can further have

$$\|\mathbf{y}_{t+1,\parallel}\| \geq (1 + \sqrt{\rho\epsilon}/(2\ell) - \rho r/(\alpha_t\ell))\|\mathbf{y}_{t,\parallel}\|. \quad (17)$$

Intuitively, $\|\mathbf{y}_{t,\parallel}\|$ should have a faster increasing rate than $\|\mathbf{y}_{t,\perp}\|$ in this gradient-based Hessian power method, ignoring the deviation from quadratic approximation. As a result, the value the value $\alpha_t = \|\mathbf{y}_{t,\parallel}\|/\|\mathbf{y}_t\|$ should be non-decreasing. It is demonstrated in Lemma 17 in Appendix B that, even if we count this deviation in, $\alpha_t$ can still be lower bounded by some constant $\alpha_{\min}$:

$$\alpha_t \geq \alpha_{\min} = \frac{\delta_0}{4}\sqrt{\frac{\pi}{n}}, \qquad \forall 1 \leq t \leq \mathscr{T}. \quad (18)$$

by which we can further deduce that

$$\|\mathbf{y}_{t,\parallel}\| \geq \|\mathbf{y}_{0,\parallel}\|(1 + \sqrt{\rho\epsilon}/\ell - \rho r/(\alpha_{\min}\ell))^t \geq \|\mathbf{y}_{0,\parallel}\|(1 + 7\sqrt{\rho\epsilon}/(8\ell))^t, \quad \forall 1 \leq t \leq \mathscr{T}. \quad (19)$$

Observe that

$$\frac{\|\mathbf{y}_{\mathscr{T},\perp'}\|}{\|\mathbf{y}_{\mathscr{T},\parallel}\|} \leq \frac{\|\mathbf{y}_{0,\perp'}\|}{\|\mathbf{y}_{0,\parallel}\|} \cdot \left(\frac{1 + 5\sqrt{\rho\epsilon}/(8\ell)}{1 + 7\sqrt{\rho\epsilon}/(8\ell)}\right)^{\mathscr{T}} \leq \frac{1}{\delta_0}\sqrt{\frac{n}{\pi}}\left(\frac{1 + 5\sqrt{\rho\epsilon}/(8\ell)}{1 + 7\sqrt{\rho\epsilon}/(8\ell)}\right)^{\mathscr{T}} \leq \frac{\sqrt{\rho\epsilon}}{8\ell}. \quad (20)$$

Since $\|\mathbf{y}_{\mathscr{T},\parallel}\| \leq \|\mathbf{y}_\mathscr{T}\|$, we have $\|\mathbf{y}_{\mathscr{T},\perp'}\|/\|\mathbf{y}_\mathscr{T}\| \leq \sqrt{\rho\epsilon}/(8\ell)$, contradiction. Hence, there here exists some $t_0$ with $1 \leq t_0 \leq \mathscr{T}$ such that $\|\mathbf{y}_{t_0,\perp'}\|/\|\mathbf{y}_{t_0}\| \leq \sqrt{\rho\epsilon}/(8\ell)$. Consider the normalized vector $\hat{\mathbf{e}} = \mathbf{y}_{t_0}/r$, we use $\hat{\mathbf{e}}_{\perp'}$ and $\hat{\mathbf{e}}_{\parallel'}$ to separately denote the component of $\hat{\mathbf{e}}$ in $\mathfrak{S}'_\perp$ and $\mathfrak{S}'_\parallel$. Then, $\|\hat{\mathbf{e}}_{\perp'}\| \leq \sqrt{\rho\epsilon}/(8\ell)$ whereas $\|\hat{\mathbf{e}}_{\parallel'}\| \geq 1 - \rho\epsilon/(8\ell)^2$. Then,

$$\hat{\mathbf{e}}^T\mathcal{H}(0)\hat{\mathbf{e}} = (\hat{\mathbf{e}}_{\perp'} + \hat{\mathbf{e}}_{\parallel'})^T\mathcal{H}(0)(\hat{\mathbf{e}}_{\perp'} + \hat{\mathbf{e}}_{\parallel'}) = \hat{\mathbf{e}}_{\perp'}^T\mathcal{H}(0)\hat{\mathbf{e}}_{\perp'} + \hat{\mathbf{e}}_{\parallel'}^T\mathcal{H}(0)\hat{\mathbf{e}}_{\parallel'}, \quad (21)$$

since $\mathcal{H}(0)\hat{\mathbf{e}}_{\perp'} \in \mathfrak{S}'_\perp$ and $\mathcal{H}(0)\hat{\mathbf{e}}_{\parallel'} \in \mathfrak{S}'_\parallel$. Due to the $\ell$-smoothness of the function, all eigenvalue of the Hessian matrix has its absolute value upper bounded by $\ell$. Hence,

$$\hat{\mathbf{e}}_{\perp'}^T\mathcal{H}(0)\hat{\mathbf{e}}_{\perp'} \leq \ell\|\hat{\mathbf{e}}_{\perp'}^T\|_2^2 = \rho\epsilon/(64\ell^2). \quad (22)$$

Further according to the definition of $\mathfrak{S}_\parallel$, we have

$$\hat{\mathbf{e}}_{\parallel'}^T\mathcal{H}(0)\hat{\mathbf{e}}_{\parallel'} \leq -\sqrt{\rho\epsilon}\|\hat{\mathbf{e}}_{\parallel'}\|^2/2. \quad (23)$$

Combining these two inequalities together, we can obtain

$$\hat{\mathbf{e}}^T\mathcal{H}(0)\hat{\mathbf{e}} = \hat{\mathbf{e}}_\perp^T\mathcal{H}(0)\hat{\mathbf{e}}_{\perp'} + \hat{\mathbf{e}}_{\parallel'}^T\mathcal{H}(0)\hat{\mathbf{e}}_{\parallel'} \leq -\sqrt{\rho\epsilon}\|\hat{\mathbf{e}}_{\parallel'}\|^2/2 + \rho\epsilon/(64\ell^2) \leq -\sqrt{\rho\epsilon}/4. \quad (24)$$

$\square$

**Remark 4.** *In practice, the value of $\|\mathbf{y}_t\|$ can become large during the execution of Algorithm 1. To fix this, we can renormalize $\mathbf{y}_t$ to have $\ell_2$-norm $r$ at the ends of such iterations, and this does not influence the performance of the algorithm.*

## 2.2 Faster negative curvature finding based on accelerated gradient descents

In this subsection, we replace the GD part in Algorithm 1 by AGD to obtain an accelerated negative curvature finding subroutine with similar effect and faster convergence rate, based on which we further implement our Accelerated Gradient Descent with Negative Curvature Finding (Algorithm 2). Near any saddle point $\tilde{\mathbf{x}} \in \mathbb{R}^n$ with $\lambda_{\min}(\mathcal{H}(\tilde{\mathbf{x}})) \leq -\sqrt{\rho\epsilon}$, Algorithm 2 finds a unit vector $\hat{\mathbf{e}}$ such that $\hat{\mathbf{e}}^T \mathcal{H}(\tilde{\mathbf{x}})\hat{\mathbf{e}} \leq -\sqrt{\rho\epsilon}/4$.

---

**Algorithm 2:** Perturbed Accelerated Gradient Descent with Accelerated Negative Curvature Finding$(\mathbf{x}_0, \eta, \theta, \gamma, s, \mathscr{T}', r')$

---

**1** $t_{\text{perturb}} \leftarrow 0, \mathbf{z}_0 \leftarrow \mathbf{x}_0, \tilde{\mathbf{x}} \leftarrow \mathbf{x}_0, \zeta \leftarrow 0$;
**2** **for** $t = 0, 1, 2, ..., T$ **do**
**3**    **if** $\|\nabla f(\mathbf{x}_t)\| \leq \epsilon$ *and* $t - t_{perturb} > \mathscr{T}$ **then**
**4**      $\tilde{\mathbf{x}} = \mathbf{x}_t$;
**5**      $\mathbf{x}_t \leftarrow \text{Uniform}(\mathbb{B}_{\tilde{\mathbf{x}}}(r'))$ where $\text{Uniform}(\mathbb{B}_{\tilde{\mathbf{x}}}(r'))$ is the $\ell_2$-norm ball centered at $\tilde{\mathbf{x}}$ with radius $r'$, $\mathbf{z}_t \leftarrow \mathbf{x}_t, \zeta \leftarrow \nabla f(\tilde{\mathbf{x}}), t_{\text{perturb}} \leftarrow t$;
**6**    **if** $t - t_{perturb} = \mathscr{T}'$ **then**
**7**      $\hat{\mathbf{e}} := \frac{\mathbf{x}_t - \tilde{\mathbf{x}}}{\|\mathbf{x}_t - \tilde{\mathbf{x}}\|}$;
**8**      $\mathbf{x}_t \leftarrow \tilde{\mathbf{x}} - \frac{f_{\hat{\mathbf{e}}}'(\tilde{\mathbf{x}})}{4|f_{\hat{\mathbf{e}}}'(\tilde{\mathbf{x}})|}\sqrt{\frac{\epsilon}{\rho}} \cdot \hat{\mathbf{e}}, \mathbf{z}_t \leftarrow \mathbf{x}_t, \zeta = \mathbf{0}$;
**9**    $\mathbf{x}_{t+1} \leftarrow \mathbf{z}_t - \eta(\nabla f(\mathbf{z}_t) - \zeta)$;
**10**    $\mathbf{v}_{t+1} \leftarrow \mathbf{x}_{t+1} - \mathbf{x}_t$;
**11**    $\mathbf{z}_{t+1} \leftarrow \mathbf{x}_{t+1} + (1 - \theta)\mathbf{v}_{t+1}$;
**12**    **if** $t_{perturb} \neq 0$ *and* $t - t_{perturb} < \mathscr{T}'$ **then**
**13**      $\mathbf{z}_{t+1} \leftarrow \tilde{\mathbf{x}} + r' \cdot \frac{\mathbf{z}_{t+1} - \tilde{\mathbf{x}}}{\|\mathbf{z}_{t+1} - \tilde{\mathbf{x}}\|}, \mathbf{x}_{t+1} \leftarrow \tilde{\mathbf{x}} + r' \cdot \frac{\mathbf{x}_{t+1} - \tilde{\mathbf{x}}}{\|\mathbf{z}_{t+1} - \tilde{\mathbf{x}}\|}$;
**14**    **else**
**15**      **if** $f(\mathbf{x}_{t+1}) \leq f(\mathbf{z}_{t+1}) + \langle \nabla f(\mathbf{z}_{t+1}), \mathbf{x}_{t+1} - \mathbf{z}_{t+1}\rangle - \frac{\gamma}{2}\|\mathbf{z}_{t+1} - \mathbf{x}_{t+1}\|^2$ **then**
**16**        $(\mathbf{x}_{t+1}, \mathbf{v}_{t+1}) \leftarrow \text{NegativeCurvatureExploitation}(\mathbf{x}_{t+1}, \mathbf{v}_{t+1}, s)^3$;
**17**      $\mathbf{z}_{t+1} \leftarrow \mathbf{x}_{t+1} + (1 - \theta)\mathbf{v}_{t+1}$;

---

The following proposition exhibits the effectiveness of Algorithm 2 for finding negative curvatures near saddle points:

**Proposition 5.** *Suppose the function $f \colon \mathbb{R}^n \to \mathbb{R}$ is $\ell$-smooth and $\rho$-Hessian Lipschitz. For any $0 < \delta_0 \leq 1$, we specify our choice of parameters and constants we use as follows:*

$$\eta := \frac{1}{4\ell} \qquad \theta := \frac{(\rho\epsilon)^{1/4}}{4\sqrt{\ell}} \qquad \mathscr{T}' := \frac{32\sqrt{\ell}}{(\rho\epsilon)^{1/4}}\log\left(\frac{\ell}{\delta_0}\sqrt{\frac{n}{\rho\epsilon}}\right)$$

$$\gamma := \frac{\theta^2}{\eta} \qquad s := \frac{\gamma}{4\rho} \qquad r' := \frac{\delta_0\epsilon}{32}\sqrt{\frac{\pi}{\rho n}} \qquad (25)$$

*then for a point $\tilde{\mathbf{x}}$ satisfying $\lambda_{\min}(\nabla^2 f(\tilde{\mathbf{x}})) \leq -\sqrt{\rho\epsilon}$, if running Algorithm 2 with the uniform perturbation in Line 5 being added at $t = 0$, the unit vector $\hat{\mathbf{e}}$ in Line 7 obtained after $\mathscr{T}'$ iterations satisfies:*

$$\mathbb{P}\left(\hat{\mathbf{e}}^T \mathcal{H}(\mathbf{x})\hat{\mathbf{e}} \leq -\sqrt{\rho\epsilon}/4\right) \geq 1 - \delta_0. \qquad (26)$$

The proof of Proposition 5 is similar to the proof of Proposition 3, and is deferred to Appendix B.2.

## 2.3 Escaping saddle points using negative curvature finding

In this subsection, we demonstrate that our Algorithm 1 and Algorithm 2 with the ability to find negative curvature near saddle points can further escape saddle points of nonconvex functions. The

---

[3] This NegativeCurvatureExploitation (NCE) subroutine was originally introduced in [22, Algorithm 3] and is called when we detect that the current momentum $\mathbf{v}_t$ coincides with a negative curvature direction of $\mathbf{z}_t$. In this case, we reset the momentum $\mathbf{v}_t$ and decide whether to exploit this direction based on the value of $\|\mathbf{v}_t\|$.

intuition works as follows: we start with gradient descents or accelerated gradient descents until the gradient becomes small. At this position, we compute the negative curvature direction, described by a unit vector $\hat{\mathbf{e}}$, via Algorithm 1 or the negative curvature finding subroutine of Algorithm 2. Then, we add a perturbation along this direction of negative curvature and go back to gradient descents or accelerated gradient descents with an additional NegativeCurvatureExploitation subroutine (see Footnote 3). It has the following guarantee:

**Lemma 6.** *Suppose the function $f : \mathbb{R}^n \to \mathbb{R}$ is $\ell$-smooth and $\rho$-Hessian Lipschitz. Then for any point $\mathbf{x}_0 \in \mathbb{R}^n$, if there exists a unit vector $\hat{\mathbf{e}}$ satisfying $\hat{\mathbf{e}}^T \mathcal{H}(\mathbf{x}_0) \hat{\mathbf{e}} \leq -\frac{\sqrt{\rho\epsilon}}{4}$ where $\mathcal{H}$ stands for the Hessian matrix of function $f$, the following inequality holds:*

$$f\Big(\mathbf{x}_0 - \frac{f'_{\hat{\mathbf{e}}}(\mathbf{x}_0)}{4|f'_{\hat{\mathbf{e}}}(\mathbf{x}_0)|}\sqrt{\frac{\epsilon}{\rho}} \cdot \hat{\mathbf{e}}\Big) \leq f(\mathbf{x}_0) - \frac{1}{384}\sqrt{\frac{\epsilon^3}{\rho}}, \tag{27}$$

*where $f'_{\hat{\mathbf{e}}}$ stands for the gradient component of $f$ along the direction of $\hat{\mathbf{e}}$.*

*Proof.* Without loss of generality, we assume $\mathbf{x}_0 = \mathbf{0}$. We can also assume $\langle \nabla f(\mathbf{0}), \hat{\mathbf{e}} \rangle \leq 0$; if this is not the case we can pick $-\hat{\mathbf{e}}$ instead, which still satisfies $(-\hat{\mathbf{e}})^T \mathcal{H}(\mathbf{x}_0)(-\hat{\mathbf{e}}) \leq -\frac{\sqrt{\rho\epsilon}}{4}$. In practice, to figure out whether we should use $\hat{\mathbf{e}}$ or $-\hat{\mathbf{e}}$, we apply both of them in (27) and choose the one with smaller function value. Then, for any $\mathbf{x} = x_{\hat{\mathbf{e}}}\hat{\mathbf{e}}$ with some $x_{\hat{\mathbf{e}}} > 0$, we have $\frac{\partial^2 f}{\partial x_{\hat{\mathbf{e}}}^2}(\mathbf{x}) \leq -\frac{\sqrt{\rho\epsilon}}{4} + \rho x_{\hat{\mathbf{e}}}$ due to the $\rho$-Hessian Lipschitz condition of $f$. Hence,

$$\frac{\partial f}{\partial x_{\hat{\mathbf{e}}}}(\mathbf{x}) \leq f'_{\hat{\mathbf{e}}}(\mathbf{0}) - \frac{\sqrt{\rho\epsilon}}{4}x_{\hat{\mathbf{e}}} + \rho x_{\hat{\mathbf{e}}}^2, \tag{28}$$

by which we can further derive that

$$f(x_{\hat{\mathbf{e}}}\hat{\mathbf{e}}) - f(\mathbf{0}) \leq f'_{\hat{\mathbf{e}}}(\mathbf{0})x_{\hat{\mathbf{e}}} - \frac{\sqrt{\rho\epsilon}}{8}x_{\hat{\mathbf{e}}}^2 + \frac{\rho}{3}x_{\hat{\mathbf{e}}}^3 \leq -\frac{\sqrt{\rho\epsilon}}{8}x_{\hat{\mathbf{e}}}^2 + \frac{\rho}{3}x_{\hat{\mathbf{e}}}^3. \tag{29}$$

Settings $x_{\hat{\mathbf{e}}} = \frac{1}{4}\sqrt{\frac{\epsilon}{\rho}}$ gives (27). $\qquad\square$

We give the full algorithm details based on Algorithm 1 in Appendix C.1. Along this approach, we achieve the following:

**Theorem 7** (informal, full version deferred to Appendix C.3)**.** *For any $\epsilon > 0$ and a constant $0 < \delta \leq 1$, Algorithm 2 satisfies that at least one of the iterations $\mathbf{x}_t$ will be an $\epsilon$-approximate second-order stationary point in*

$$\tilde{O}\Big(\frac{(f(\mathbf{x}_0) - f^*)}{\epsilon^{1.75}} \cdot \log n\Big) \tag{30}$$

*iterations, with probability at least $1 - \delta$, where $f^*$ is the global minimum of $f$.*

Intuitively, the proof of Theorem 7 has two parts. The first part is similar to the proof of [22, Theorem 3], which shows that PAGD uses $\tilde{O}(\log^6 n/\epsilon^{1.75})$ iterations to escape saddle points. We show that there can be at most $\tilde{O}(\Delta_f/\epsilon^{1.75})$ iterations with the norm of gradient larger than $\epsilon$ using almost the same techniques, but with slightly different parameter choices. The second part is based on the negative curvature part of Algorithm 2, our accelerated negative curvature finding algorithm. Specifically, at each saddle point we encounter, we can take $\tilde{O}(\log n/\epsilon^{1/4})$ iterations to find its negative curvature (Proposition 5), and add a perturbation in this direction to decrease the function value by $O(\epsilon^{1.5})$ (Lemma 6). Hence, the iterations introduced by Algorithm 4 can be at most $\tilde{O}\big(\frac{\log n}{\epsilon^{1.5}} \cdot \frac{1}{\epsilon^{0.25}}\big) = \tilde{O}(\log n/\epsilon^{1.75})$, which is simply an upper bound on the overall iteration number. The detailed proof is deferred to Appendix C.3.

**Remark 8.** *Although Theorem 7 only demonstrates that our algorithms will visit some $\epsilon$-approximate second-order stationary point during their execution with high probability, it is straightforward to identify one of them if we add a termination condition: once Negative Curvature Finding (Algorithm 1 or Algorithm 2) is applied, we record the position $\mathbf{x}_{t_0}$ and the function value decrease due to the following perturbation. If the function value decrease is larger than $\frac{1}{384}\sqrt{\frac{\epsilon^3}{\rho}}$ as per Lemma 6, then the algorithms make progress. Otherwise, $\mathbf{x}_{t_0}$ is an $\epsilon$-approximate second-order stationary point with high probability.*

# 3 Stochastic setting

In this section, we present a stochastic version of Algorithm 1 using stochastic gradients, and demonstrate that it can also be used to escape saddle points and obtain a polynomial speedup in $\log n$ compared to the perturbed stochastic gradient (PSGD) algorithm in [21].

## 3.1 Stochastic negative curvature finding

In the stochastic gradient descent setting, the exact gradients oracle $\nabla f$ of function $f$ cannot be accessed. Instead, we only have unbiased stochastic gradients $\mathbf{g}(\mathbf{x}; \theta)$ such that

$$\nabla f(\mathbf{x}) = \mathbb{E}_{\theta \sim \mathcal{D}}[\mathbf{g}(\mathbf{x}; \theta)] \qquad \forall \mathbf{x} \in \mathbb{R}^n, \tag{31}$$

where $\mathcal{D}$ stands for the probability distribution followed by the random variable $\theta$. Define

$$\zeta(\mathbf{x}; \theta) := g(\mathbf{x}; \theta) - \nabla f(\mathbf{x}) \tag{32}$$

to be the error term of the stochastic gradient. Then, the expected value of vector $\zeta(\mathbf{x}; \theta)$ at any $\mathbf{x} \in \mathbb{R}^n$ equals to $\mathbf{0}$. Further, we assume the stochastic gradient $\mathbf{g}(\mathbf{x}, \theta)$ also satisfies the following assumptions, which were also adopted in previous literatures; see e.g. [9, 20, 21, 35].

**Assumption 1.** *For any $\mathbf{x} \in \mathbb{R}^n$, the stochastic gradient $\mathbf{g}(\mathbf{x}; \theta)$ with $\theta \sim \mathcal{D}$ satisfies:*

$$\Pr[(\|\mathbf{g}(\mathbf{x}; \theta) - \nabla f(\mathbf{x})\| \geq t)] \leq 2 \exp(-t^2/(2\sigma^2)), \quad \forall t \in \mathbb{R}. \tag{33}$$

**Assumption 2.** *For any $\theta \in supp(\mathcal{D})$, $\mathbf{g}(\mathbf{x}; \theta)$ is $\tilde{\ell}$-Lipschitz for some constant $\tilde{\ell}$:*

$$\|\mathbf{g}(\mathbf{x}_1; \theta) - \mathbf{g}(\mathbf{x}_2; \theta)\| \leq \tilde{\ell}\|\mathbf{x}_1 - \mathbf{x}_2\|, \qquad \forall \mathbf{x}_1, \mathbf{x}_2 \in \mathbb{R}^n. \tag{34}$$

Assumption 2 emerges from the fact that the stochastic gradient $\mathbf{g}$ is often obtained as an exact gradient of some smooth function,

$$\mathbf{g}(\mathbf{x}; \theta) = \nabla f(\mathbf{x}; \theta). \tag{35}$$

In this case, Assumption 2 guarantees that for any $\theta \sim \mathcal{D}$, the spectral norm of the Hessian of $f(\mathbf{x}; \theta)$ is upper bounded by $\tilde{\ell}$. Under these two assumptions, we can construct the stochastic version of Algorithm 1, as shown in Algorithm 3.

---

**Algorithm 3:** Stochastic Negative Curvature Finding($\mathbf{x}_0, r_s, \mathscr{T}_s, m$).

---

1   $\mathbf{y}_0 \leftarrow 0$, $L_0 \leftarrow r_s$;
2   **for** $t = 1, ..., \mathscr{T}_s$ **do**
3     Sample $\{\theta^{(1)}, \theta^{(2)}, \cdots, \theta^{(m)}\} \sim \mathcal{D}$;
4     $\mathbf{g}(\mathbf{y}_{t-1}) \leftarrow \frac{1}{m} \sum_{j=1}^{m} (\mathbf{g}(\mathbf{x}_0 + \mathbf{y}_{t-1}; \theta^{(j)}) - \mathbf{g}(\mathbf{x}_0; \theta^{(j)}))$;
5     $\mathbf{y}_t \leftarrow \mathbf{y}_{t-1} - \frac{1}{\ell}(\mathbf{g}(\mathbf{y}_{t-1}) + \xi_t/L_{t-1})$,      $\xi_t \sim \mathcal{N}(0, \frac{r_s^2}{d}I)$;
6     $L_t \leftarrow \frac{\|\mathbf{y}_t\|}{r_s} L_{t-1}$ and $\mathbf{y}_t \leftarrow \mathbf{y}_t \cdot \frac{r_s}{\|\mathbf{y}_t\|}$;
7   **Output** $\mathbf{y}_{\mathscr{T}}/r_s$.

---

Similar to the non-stochastic setting, Algorithm 3 can be used to escape from saddle points and obtain a polynomial speedup in $\log n$ compared to PSGD algorithm in [21]. This is quantitatively shown in the following theorem:

**Theorem 9** (informal, full version deferred to Appendix D.2). *For any $\epsilon > 0$ and a constant $0 < \delta \leq 1$, our algorithm[4] based on Algorithm 3 using only stochastic gradient descent satisfies that at least one of the iterations $\mathbf{x}_t$ will be an $\epsilon$-approximate second-order stationary point in*

$$\tilde{O}\left(\frac{(f(\mathbf{x}_0) - f^*)}{\epsilon^4} \cdot \log^2 n\right) \tag{36}$$

*iterations, with probability at least $1 - \delta$, where $f^*$ is the global minimum of $f$.*

---

[4]Our algorithm based on Algorithm 3 has similarities to the Neon2[online] algorithm in [3]. Both algorithms find a second-order stationary point for stochastic optimization in $\tilde{O}(\log^2 n/\epsilon^4)$ iterations, and we both apply directed perturbations based on the results of negative curvature finding. Nevertheless, our algorithm enjoys simplicity by only having a single loop, whereas Neon2[online] has a nested loop for boosting their confidence.

# 4 Numerical experiments

In this section, we provide numerical results that exhibit the power of our negative curvature finding algorithm for escaping saddle points. More experimental results can be found in Appendix E. All the experiments are performed on MATLAB R2019b on a computer with Six-Core Intel Core i7 processor and 16GB memory, and their codes are given in the supplementary material.

**Comparison between Algorithm 1 and PGD.** We compare the performance of our Algorithm 1 with the perturbed gradient descent (PGD) algorithm in [21] on a test function $f(x_1, x_2) = \frac{1}{16}x_1^4 - \frac{1}{2}x_1^2 + \frac{9}{8}x_2^2$ with a saddle point at $(0, 0)$. The advantage of Algorithm 1 is illustrated in Figure 1.

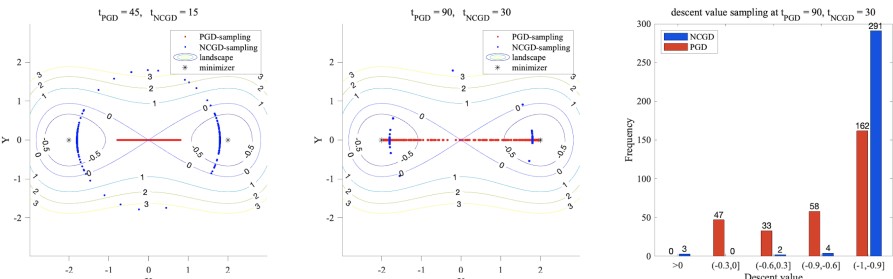

Figure 1: Run Algorithm 1 and PGD on landscape $f(x_1, x_2) = \frac{1}{16}x_1^4 - \frac{1}{2}x_1^2 + \frac{9}{8}x_2^2$. Parameters: $\eta = 0.05$ (step length), $r = 0.1$ (ball radius in PGD and parameter $r$ in Algorithm 1), $M = 300$ (number of samplings). **Left**: The contour of the landscape is placed on the background with labels being function values. Blue points represent samplings of Algorithm 1 at time step $t_{\text{NCGD}} = 15$ and $t_{\text{NCGD}} = 30$, and red points represent samplings of PGD at time step $t_{\text{PGD}} = 45$ and $t_{\text{PGD}} = 90$. Algorithm 1 transforms an initial uniform-circle distribution into a distribution concentrating on two points indicating negative curvature, and these two figures represent intermediate states of this process. It converges faster than PGD even when $t_{\text{NCGD}} \ll t_{\text{PGD}}$.
**Right**: A histogram of descent values obtained by Algorithm 1 and PGD, respectively. Set $t_{\text{NCGD}} = 30$ and $t_{\text{PGD}} = 90$. Although we run three times of iterations in PGD, there are still over $40\%$ of gradient descent paths with function value decrease no greater than 0.9, while this ratio for Algorithm 1 is less than $5\%$.

**Comparison between Algorithm 3 and PSGD.** We compare the performance of our Algorithm 3 with the perturbed stochastic gradient descent (PSGD) algorithm in [21] on a test function $f(x_1, x_2) = (x_1^3 - x_2^3)/2 - 3x_1x_2 + (x_1^2 + x_2^2)^2/2$. Compared to the landscape of the previous experiment, this function is more deflated from a quadratic field due to the cubic terms. Nevertheless, Algorithm 3 still possesses a notable advantage compared to PSGD as demonstrated in Figure 2.

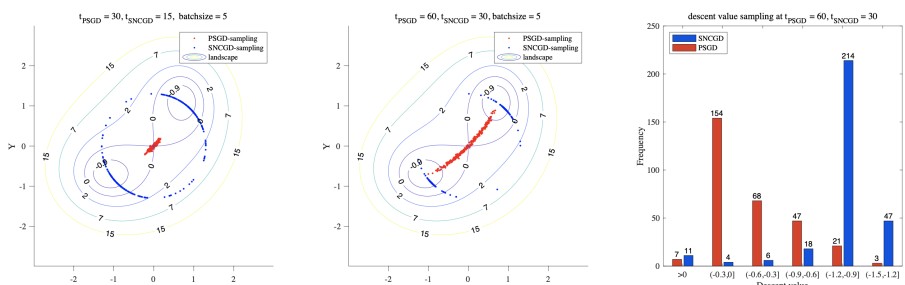

Figure 2: Run Algorithm 3 and PSGD on landscape $f(x_1, x_2) = \frac{x_1^3 - x_2^3}{2} - 3x_1x_2 + \frac{1}{2}(x_1^2 + x_2^2)^2$. Parameters: $\eta = 0.02$ (step length), $r = 0.01$ (variance in PSGD and $r_s$ in Algorithm 3), $M = 300$ (number of samplings). **Left**: The contour of the landscape is placed on the background with labels being function values. Blue points represent samplings of Algorithm 3 at time step $t_{\text{SNCGD}} = 15$ and $t_{\text{SNCGD}} = 30$, and red points represent samplings of PSGD at time step $t_{\text{PSGD}} = 30$ and $t_{\text{PSGD}} = 60$. Algorithm 3 transforms an initial uniform-circle distribution into a distribution concentrating on two points indicating negative curvature, and these two figures represent intermediate states of this process. It converges faster than PSGD even when $t_{\text{SNCGD}} \ll t_{\text{PSGD}}$.
**Right**: A histogram of descent values obtained by Algorithm 3 and PSGD, respectively. Set $t_{\text{SNCGD}} = 30$ and $t_{\text{PSGD}} = 60$. Although we run two times of iterations in PSGD, there are still over $50\%$ of SGD paths with function value decrease no greater than 0.6, while this ratio for Algorithm 3 is less than $10\%$.

## Acknowledgements

We thank Jiaqi Leng for valuable suggestions and generous help on the design of numerical experiments. TL was supported by the NSF grant PHY-1818914 and a Samsung Advanced Institute of Technology Global Research Partnership.

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
