# OpenReview forum: "Escape saddle points by a simple gradient-descent based algorithm"
_NeurIPS.cc/2021/Conference — NeurIPS 2021 Poster_

### Official Review · Reviewer_YK72 · 2021-07-16

**Rating:** 6
**Confidence:** 3

**Summary:**

In this paper, the authors propose a simple gradient-based algorithm for a smooth function. And it outputs an \epsilon-approximate second-order stationary point with the complexity O(\log n/\epsilon^1.75), which is polynomially better in terms of log(n) and matches their complexities in terms of 1/\epsilon.

**Limitations And Societal Impact:**

1.	The main contribution of this paper is an idea of implementing a robust Hessian power method using only gradients, which can find negative curvature near saddle points and achieve a polynomial speedup in log(n) compared to the perturbed gradient descent methods. However, the authors should add some comparisons with existing works based on negative curvature descent such as [26] and [R1] in both theory and practical.
2.	The author should explain the main differences between the proposed algorithm and the related works [26] and [R1].

3.	In the experiment section, the compared algorithms are too few. Can the authors provide more comparisons to other advanced optimization algorithms?

4.	The experiments are done only for the special non-convex functions, and the authors should provide more results for more general non-convex fucntions.
5.	In Line 165, it may be \alpha_t=\|y_{t,\|}\|/\|y_t\| instead of \alpha_t=\|y\|_{t,\|}/\|y_t\|.

[R1] Mingrui Liu, Zhe Li, Xiaoyu Wang, Jinfeng Yi, Tianbao Yang.  Adaptive Negative Curvature Descent with Applications in Non-convex Optimization.  NeurIPS 2018

----------after feedback---------------

The authors address my comments well, so I increased my score.

**Main Review:**

This paper is technically sound, and the experiments also seem convincing. However, this paper requires some improvements.
1.	The main contribution of this paper is an idea of implementing a robust Hessian power method using only gradients, which can find negative curvature near saddle points and achieve a polynomial speedup in log(n) compared to the perturbed gradient descent methods. However, the authors should add some comparisons with existing works based on negative curvature descent such as [26] and [R1] in both theory and practical.
2.	The author should explain the main differences between the proposed algorithm and the related works [26] and [R1].

3.	In the experiment section, the compared algorithms are too few. Can the authors provide more comparisons to other advanced optimization algorithms?

4.	The experiments are done only for the special non-convex functions, and the authors should provide more results for more general non-convex fucntions.
5.	In Line 165, it may be \alpha_t=\|y_{t,\|}\|/\|y_t\| instead of \alpha_t=\|y\|_{t,\|}/\|y_t\|.

[R1] Mingrui Liu, Zhe Li, Xiaoyu Wang, Jinfeng Yi, Tianbao Yang.  Adaptive Negative Curvature Descent with Applications in Non-convex Optimization.  NeurIPS 2018

**Time Spent Reviewing:**

72

---

> ### Author Response · Authors · 2021-08-10
> **Reply to Reviewer YK72**
>
> We appreciate for your detailed comments and suggestions!
>
> Regarding Refs. [26] and [R1], we agree that we should have given a more detailed comparison between our result and those works. As one of the earliest works investigating the performance of accelerated methods in nonconvex functions, Ref. [26] proved that accelerated methods can find a second-order stationary point in $\tilde{O}(1/\epsilon^{1.75})$ iterations using only first-order oracle. This idea was further exploited in [R1] with some novel observations that relaxes the requirement on noise level to reduce the iteration complexity, and the neat idea of encouraging a competition between a negative curvature descent and the gradient descent to achieve a maximal decrease at each step.
>
> The underlying dynamics of our work is similar to those of [26] and [R1], but the main focuses are different. Specifically, [26] and [R1] mainly aim at using novel techniques to reduce the iteration complexity for finding a second-order stationary point, whereas our work mainly focuses on reducing the number of loops and hyper-parameters of negative curvature finding methods while preserving their advantage in iteration complexity, since a much simpler structure accords with empirical observations and enables much wider applications. Moreover, we added renormalization steps in our algorithms, ensuring the numerical stability in practical situations. We will add more discussions in the final version.
>
> Regarding your suggestions on numerical experiments about including more advanced optimization algorithms and more general nonconvex landscapes, we totally agree with you on their importance. Hence, we further perform numerical experiments at https://drive.google.com/file/d/1_n3vtMTcJNyh5uvvyz46H0UKpcrSswMc/view?usp=sharing (an anonymous PDF file) to justify that in other landscapes of more general nonconvex functions, our Algorithm 2 still outperforms PAGD [19].
>
> Regarding your comment on other minor issues, we think they are very helpful and can make our paper more organized. We are happy to make corresponding changes in the final version.

---

### Official Review · Reviewer_ub9p · 2021-07-16

**Rating:** 8
**Confidence:** 3

**Summary:**

This paper provides an improved method for computing stationary points of nonconvex functions. The authors give an algorithm which, given an $L$-smooth $\rho$-Hessian Lipschitz nonconvex function $f$, computes an $(O(\epsilon), O(\sqrt{\rho \epsilon}))$-second order stationary point of $f$ using $O(\eps^{-7/4} \log n)$ gradient computations of $f$. This matches the $\epsilon$ dependence  achieved by several previous works in the area (for example [6,7,20]) but  improves the dependence on $\log n$, where $n$ is the dimension, by polynomial factors. The algorithm is also of similar simplicity to the previous state-of-the-art method given in [20], and combines the perturbed AGD framework given there with some clever observations to achieve the result.

**Limitations And Societal Impact:**

While the result obtained is nice and of obvious practical importance, the analysis of acceleration seems to be heavily borrowed from [20], outside of the clever improvement to the negative curvature exploitation. In addition, the experiments only benchmark solution quality and iteration complexity against an unaccelerated method (PGD from [19])-- it would be interesting to see if the claimed iteration complexity translates to [20]. My score is borderline between a 7 and an 8 for these reasons: I would be happy to raise my evaluation if these concerns are addressed.

**Main Review:**

All algorithms in this line of work combine accelerated gradient descent with a mechanism to exploit negative curvature in the objective. The main contribution of this work is a clever interpretation of perturbed gradient descent itself as a curvature exploitation routine: this significantly simplifies the algorithm and allows their approach to obtain results for stochastic gradient oracles as well.

The analysis of the algorithm is relatively straightforward and easy to follow. The proof of the negative curvature exploitation routine seems fully correct to me, and in my brief skim I did not see any issues with the combination of this with the acceleration framework of [20]. The paper does a good job motivating the negative curvature detection idea-- the algorithm the authors propose is both the simplest and fastest algorithm in the literature for finding stationary points of nonconvex functions.

**Time Spent Reviewing:**

2

---

> ### Author Response · Authors · 2021-08-10
> **Reply to Reviewer ub9p**
>
> We appreciate for your positive feedback and detailed suggestions!
>
> Regarding your suggestions on numerical experiments, especially about benchmarking the solution quality and iteration complexity of our algorithm against the accelerated method in [20], we totally agree with its importance. Actually, in the submitted version, we performed an experiment comparing our Algorithm 2 with the PAGD from [20] in Appendix E of the supplementary material. Nevertheless, we did further numerical experiments at https://drive.google.com/file/d/1_n3vtMTcJNyh5uvvyz46H0UKpcrSswMc/view?usp=sharing (an anonymous PDF file) to justify that compared to PAGD and even advanced optimization algorithms, Algorithm 2 possesses better solution quality and iteration complexity in various landscapes.
>
> Regarding your comment on the similarities between our analysis and the acceleration framework of [20], we acknowledge that our Algorithm 2 is the same as the PAGD algorithm [20] in large gradient regions. Current algorithms for finding second-order stationary points basically consist of two parts: the large gradient part and the escaping from saddle point part. The former part in general contributes to the dominating term in the iteration complexity, and our contribution mainly lies in the latter part when the gradient is small to improve the iteration complexity. For the formal part, we directly adopted the technique from [20], as it does not affect the overall bound. We will add more discussions in the final version.

---

### Official Review · Reviewer_Ff6E · 2021-07-17

**Rating:** 6
**Confidence:** 4

**Summary:**

In this work, the authors study the escaping saddle points problem for nonconvex optimization. Previously Jin et al. proposed a perturbed accelerated gradient descent (AGD) method which finds $\epsilon$-second-order stationary points in $O(\log^6 n/\epsilon^{1.75})$ number of iterations, where $n$ is the dimension. By utilizing a refined perturbed negative curvature estimator, the authors successfully improves the dependence of dimension from $\log^6 n$ to $\log n$. Experiment results suggest that their algorithm performs well in practice.

**Limitations And Societal Impact:**

The authors discussed the limitations and there is no negative societal impact about this work.

**Main Review:**

The main contribution of this work is a robust Hessian power method. Compared with previous approaches such as Jin et al. which estimate the negative curvature direction by a random added perturbation, this work further replaces the random perturbation by the previous estimated direction, which is a more accurate estimate. The idea is neat, the presentation is clear, and the experiments are comprehensive. Here are my additional comments.

- One of the issues of this work is the so-called ‘robustness’ property. The authors name the ‘robustness’ due to the facts that replacing GD by either SGD or AGD will not prevent the algorithm from finding negative curvature. I find it a bit confusing, since normally I would use the word ‘robustness’ to describe that an algorithm is robust to some adversarial attacks brought by the opponent, which is far away from the meaning in this work.

- Another issue of this work is the difference between the proposed estimator Algorithm 1 and the Neon 2 estimator proposed by Allen-zhu and Li. It seems that these two estimators utilize the same idea, which is to estimate the negative curvature direction through the Hessian-vector product. However, Algorithm 1 estimates the negative curvature with a single loop with high probability, while Neon 2 needs an additional boosting step to achieve a similar goal. Can the authors elaborate more on the connection and difference between them?

- Finally, for the stochastic setting, some earlier works [1, 2] proposed algorithms to find second-order stationary points without a nested-loop structure. [3, 4] studied third-order smoothness assumption and show that it can help algorithms find second-order stationary points faster. [5, 6] showed that escaping from saddle points on the low-rank matrix factorization problem is identical to finding the global minima. [7] studied a stochastic variant of the cubic-regularization method and showed a theoretical guarantee. The authors may want to comment on these related works.

[1] Ge, R., Li, Z., Wang, W., & Wang, X. (2019, June). Stabilized SVRG: Simple variance reduction for nonconvex optimization. In Conference on learning theory (pp. 1394-1448). PMLR.

[2] Li, Z. (2019). SSRGD: Simple stochastic recursive gradient descent for escaping saddle points. arXiv preprint arXiv:1904.09265.

[3] Yu, Y., Xu, P., & Gu, Q. (2018). Third-order smoothness helps: Faster stochastic optimization algorithms for finding local minima. Advances in neural information processing systems.

[4] Zhou, D., Xu, P., & Gu, Q. (2020). Stochastic nested variance reduction for nonconvex optimization. Journal of machine learning research.

[5] Ge, R., Jin, C., & Zheng, Y. (2017, July). No spurious local minima in nonconvex low rank problems: A unified geometric analysis. In International Conference on Machine Learning (pp. 1233-1242). PMLR.

[6] Zhang, X., Wang, L., Yu, Y., & Gu, Q. (2018, July). A primal-dual analysis of global optimality in nonconvex low-rank matrix recovery. In International conference on machine learning (pp. 5862-5871). PMLR.

[7] Zhou, D., Xu, P., & Gu, Q. (2018, July). Stochastic variance-reduced cubic regularized Newton methods. In International Conference on Machine Learning (pp. 5990-5999). PMLR.





**Time Spent Reviewing:**

5

---

> ### Author Response · Authors · 2021-08-10
> **Reply to Reviewer Ff6E**
>
> We appreciate for your detailed comments and suggestions!
>
> Regarding your comment on our 'robustness' property, we agree that we should elaborate more on this argument. Although not presented explicitly, our proof can be easily extended to the case where adversarial attacks are present. Specifically, in our proof of Proposition 1, we essentially view the deflation from pure quadratic geometry as an external noise that our algorithm has to overcome. We will add more discussions regarding adversary attacks to support our 'robustness' claim in the final version if our paper were accepted.
>
> Regarding the NEON2 algorithm in Ref. [3], we agree that we should have given a more detailed comparison between our result and that work. We admit that the underlining dynamics of NEON2 and our work are similar. However, there exist the following important differences:
>
> - The choice of perturbation and analysis of NEON2 is based on the Chebyshev approximation theory. In the stochastic setting, our Algorithm 3 adds a perturbation at each step, while NEON2$^{\text{online}}_{\text{weak}}$ only adds a perturbation at the first step. As a result, Algorithm 3 only needs a single loop to achieve the desired success probability, whereas in NEON2$^{\text{online}}$ an additional boosting step is required to bound the failure probability, which introduces additional loop and hyper-parameters that potentially impact the simplicity of the algorithm.
>
> - Simpler structure of our algorithms enables us to perform renormalizations at each iteration, which is not feasible in NEON2. Although in theoretical aspect a renormalization step does not affect the output and the complexity of our algorithms, it enables us to choose a much larger perturbation radius and drastically reduce the numerical instability, guaranteeing the performance of our algorithms in practical situations.
>
> We will add more discussions on this in the final version.
>
> Regarding the relevant works you mentioned (in this paragraph, we cite the reference numbers following your list), we agree that we should elaborate more to illustrate the connections between these results and our paper. Specifically, although general nonconvex functions are discussed in [2], the main focus of [1,2] is nonconvex finite-sum problems, where the objective function can be expressed as the sum of $n$ functions. Hence, $n$ is contained in the overall complexity. Although in [2] a neat bound for general nonconvex functions is also provided, its SSRGD algorithm contains an epoch structure. Hence, despite its simpleness compared to other existing works, it is still nested-looped. Nevertheless, the underlying dynamics of SSRGD is intriguing and we think it is worth investigating whether it can be merged into our simple structure. In [3,4], third-order smoothness assumption is crucial and contributes to their fast convergence rate. We are not very familiar with this line of work and the exact role of third-order smoothness in these results by now due to time limitation, but we agree with you that it is interesting to investigate whether third-order smoothness can help with our negative-curvature finding procedure. Refs. [5,6] analyzed the landscape of an important problem: low-rank matrix completion and factorization, proving that all the local minima are equally good and finding any of them can solve the problem efficiently. Hence, escaping saddle points is equivalent to finding a global optimum in this landscape. These two works demonstrate the value of the escaping saddle points problem. Refs. [7] introduced a novel idea improving the convergence rate of the previous state-of-the-art cubic regularization method using variance-reduction techniques. However, a second-order oracle is required in this procedure. Thus, it is not directly comparable to our work. We will add more discussions on these mentioned works in our final version.

---

### Official Review · Reviewer_sM4q · 2021-07-26

**Rating:** 7
**Confidence:** 4

**Summary:**

The paper considers SGD and accelerated GD approaches. While their results are similar to that of Jin et al., they achieve much better iteration dependence on log n: log n for AGD and log^2 n for SGD.

**Limitations And Societal Impact:**

I believe the paper has the following shortcomings:
I’m confused what the authors mean by “single-loop”. Algorithm 6 uses Accelerated Negative Curvature Finding, which introduces a nested loop and has the same problem with hyperparameters.
I think the paper and the main body should be more self-sufficient.For example, Theorem 3 (one of the main results) references Algorithm 6, but the algorithm itself is deferred to appendix. Negative Curvature exploitation from [20] is used, but the motivation behind it is unclear, unless one checks the original paper..
Line 252:  “we can construct the stochastic version of Algorithm 1, as shown in Algorithm 3”. It’s not clear to me why Algorithm 3 is a version of Algorithm 1.

Minor issues:
Line 151 - y_{t, ||} is used everywhere
Line 208 - assumption on < ∇ f, \bar e> should be a part of the statement
I think it would be more convenient if theorems, statements and propositions had the same numeration.
Lemma 2 - I suggest simply saying for “x_{t+1} = x_t - η ∇ f(x_t)”, since the current statement is imprecise.
Lemma 9 - Please remind what G_\perp and G_|| are


**Main Review:**

Their SGD algorithm also enjoys the property of being a single-loop approach, which allows it to achieve simplicity and reduce a smaller number of hyper-parameters. One of key ideas in the paper is a power SGD-based approach which allows one to efficiently find a direction to escape from a saddle point.


**Time Spent Reviewing:**

5

---

> ### Author Response · Authors · 2021-08-10
> **Reply to Reviewer sM4q**
>
> We appreciate for your positive feedback and detailed suggestions!
>
> Regarding your comment on the nested loop issue, we agree with you that we should elaborate more on our 'single-loop' argument to avoid confusions. We claim our algorithms to be single-looped due to the following reasons. First, our Algorithm 5 and Algorithm 6 have exactly the same dynamics inside or outside the inner loop for negative curvature finding as well as the number of hyper-parameters compared to the single-looped PGD (Ref. [19]) and PAGD (Ref. [20]), respectively. Second, our algorithms can be simply transformed to single-looped forms by deleting the renormalization step, whose only role is to improve the numerical stability and does not affect the output of the algorithm. Take Algorithm 5 as an example, it has a single-looped variant with minor modifications:
>
> Algorithm: Perturbed Gradient Descent with Negative Curvature Finding (variant)
>
> Input: $x_0\in\mathbb{R}^n$
> - For t=0,1,...,T do
> - $t_{perturb}=0$;
> - if $\|\nabla f(x_t)\|\leq\epsilon$ and $t-t_{\text{perturb}}>\mathscr{T}$:
> -- $x_t\leftarrow$ Uniform$(\mathbb{B}_{x_t}(r))$;
> -- $t_{\text{perturb}}\leftarrow t$
> -- end if;
> - if $t-t_{\text{perturb}}=\mathscr{T}$:
> -- $\hat{e}=\frac{x_t-x_{t_{\text{perturb}}}}{\|x_t-x_{t_{\text{perturb}}}\|}$;
> -- $x_t \leftarrow x_{t_{\text{perturb}}}-\frac{f_{\hat{e}}'(x_0)}{4|f_{\hat{e}}'(x_{t_{\text{perturb}}})|}$$\sqrt{\frac{\epsilon}{\rho}}\cdot\hat{e}$;
> -- end if;
> - $x_{t+1}\leftarrow x_{t}-\frac{1}{\ell}\nabla f(x_{t})$;
> - end for;
>
> Similar single-looped variant of Algorithm 6 can also be constructed with minor modifications in the proof. In the submitted version, we adopted the original version of Algorithm 5 and Algorithm 6 mainly because the renormalization step enables a larger value of the perturbation radius $r$ to improve the numerical stability by a large scale, while not affecting the simplicity of our algorithms. We will add more discussions on this in the final version of our paper.
>
> Regarding your comment on the negative curvature exploitation subroutine proposed in [20], our contributions are mainly from the escaping from saddle point procedure where the gradient is small. In large gradient regions, our Algorithm 6 is the same as the PAGD algorithm [20], including the NCE subroutine on which we do not make improvements. Nevertheless, we agree with you we should add some descriptions about the motivation of NCE subroutine, and we will address this issue in our final version.
>
> Regarding your comment on the relationship between Algorithm 1 and Algorithm 3, their main difference is that Algorithm 1 only adds a perturbation in the first step, whereas Algorithm 3 adds a perturbation in every step. The variable $L_t$ in Algorithm 3 is just an auxiliary renormalization factor to ensure numerical stability. Our claim that Algorithm 3 is the stochastic variant of Algorithm 1 follows the convention from [19], where the only difference between their perturbed gradient descent (PGD) algorithm and perturbed stochastic gradient descent (PSGD) algorithm is that PGD only perturbs once during a time period while PSGD perturbs at each iteration, similar to our case.
>
> Regarding your comments on other minor issues, we appreciate them as they can make our paper more organized. We are happy to make corresponding changes in the final version if our paper were accepted.

---

### Comment · Area_Chair_Jwqu · 2021-09-25
**Comments**

I have been asked to provide an additional opinion on this paper, and would like to call the following two points to attention.

**Gradient vs. Hessian-vector product (HVP) oracles in the noiseless setting.** The paper currently claims to improve on the "state of the art" complexity bounds for "gradient based" algorithms in the setting of an exact gradient oracle and Lipschitz-Hessian functions. However, this claim is imprecise because in the above setting there is essentially no difference between a gradient oracle and an HVP oracle. More precisely, when the Hessian is Lipschitz, approximating an HVP via a difference of two gradient queries can achieve arbitrarily low error (say $\epsilon^{80}$). Consequently, any numerically-stable algorithm using an HVP oracle can be implemented with at most twice the number of calls to a gradient oracle. Therefore, the results of [1,7] already imply a rate of $\tilde{O}(\epsilon^{-7/4} \log n)$. The authors should clarify that this rate is not new in a strict oracle complexity sense, and include [1, 7] in Table 1.

**Single-loop vs. nested-loop algorithms.** It is important to note that the single-loop/nested-loop distinction is aesthetical and not mathematical, because with enough flow control one can turn any nested loop into a single loop. Therefore, the question is not really how many loops an algorithm has but rather how elegant an algorithm is to describe and implement. I am not convinced that a single-loop version of Algorithm 6 would be nearly as elegant as the AGD variant of [20], since it’s not clear how to merge the steps in Algorithm 4 with those of Algorithm 6 (e.g., one has a $\theta$ parameter and one does not). The authors should revise their paper to spell out the single-loop version of Algorithm 6, and place it in the main paper rather than the appendix.

---

> ### Author Response · Authors · 2021-09-26
> **Reply to Area Chair Jwqu**
>
> We appreciate for your detailed comments and suggestions!
>
> Regarding your first comment on related works based on the Hessian-vector product (HVP) oracle, we agree with you that if speaking in the strict oracle complexity sense, our result is not totally new. We are happy to clarify this more and modify our Table 1 correspondingly. In fact, we claim our result to improve the complexity bound for simple gradient-based algorithm following similar argument in Ref. [19], in particular, these existing works using the HVP oracle all possess nested loops in their implementation which raises concern with the setting of hyperparameters. Moreover, although in principle an HVP oracle can be implemented by differentiating the gradient of two nearby points, such implementation may suffer from numerical stability: to reduce the error, these two points have to be close enough, which amplifies the float error. As far as we see, these works have not provided analysis on this issue and specified how close these two gradient queries should be.
>
> As for your second comment on the difference between single-looped and nested-looped algorithms, we agree with you that in our current manuscript, the presentation is a bit unclear regarding how to prepare a single-loop version of Algorithm 6 that is nearly as elegant as the PAGD in Ref. [20], because the parameters including $\theta$ and the subroutine NCE are removed in our Algorithm 4. In fact, we write the current version because they are useful in the large gradient scenario but redundant for negative curvature detection; however, we can also keep them in Algorithm 4 without affecting its complexity bound, which enables Algorithm 6 to be more succinct and fit into our single-loop structure. We will change Algorithm 4 and Algorithm 6 as well as corresponding proofs in our final version to address this issue.

---

### Decision · Program_Chairs · 2021-09-27

**Decision:**

Accept (Poster)

**Comment:**

​​This paper proposes a robust Hessian power method to find negative curvature direction, based on which (stochastic) gradient descent can find the local minima with improved gradient complexity (the improvement is in the poly-log dependence in the problem dimension $n$). All the reviewers are in strong support of this paper.  I, therefore, recommend acceptance.